# On Early Extinction and the Effect of Travelling in the SIR Model

**Petra Berenbrink**[1]    **Colin Cooper**[2]    **Cristina Gava**[2]    **David Kohan Marzagão**[3]    **Frederik Mallmann-Trenn**[2]

**Tomasz Radzik**[2]

[1]Department of Informatics, Universität Hamburg, Germany
[2]Department of Informatics, King's College London
[3]Department of Engineering Science, University of Oxford

## Abstract

We consider a population protocol version of the SIR model. In every round, an individual is chosen uniformly at random. If the individual is susceptible, then it becomes infected w.p. $\beta I_t/N$, where $I_t$ is the number of infections at time $t$ and $N$ is the total number of individuals. If the individual is infected, then it recovers w.p. $\gamma$, whereas, if the individual is already recovered, nothing happens. We prove sharp bounds on the probability of the disease becoming pandemic vs extinguishing early (dying out quickly). The probability of extinguishing early, $\mathbf{Pr}\left[\mathcal{E}_{ext}\right]$, is typically neglected in prior work since most use (deterministic) differential equations. Leveraging on this, using $\mathbf{Pr}\left[\mathcal{E}_{ext}\right]$, we proceed by bounding the expected size of the population that contracts the disease $\mathbf{E}\left[R_\infty\right]$. Prior work only calculated $\mathbf{E}\left[R_\infty \mid \overline{\mathcal{E}_{ext}}\right]$, or obtained non-closed form solutions.

We then study the two-country model also accounting for the role of $\mathbf{Pr}\left[\mathcal{E}_{ext}\right]$. We assume that both countries have different infection rates $\beta^{(i)}$, but share the same recovery rate $\gamma$. In this model, each round has two steps: First, an individual is chosen u.a.r. and travels w.p. $p_{travel}$ to the other country. Afterwards, the process continues as before with the respective infection rates.

Finally, using simulations, we characterise the influence of $p_{travel}$ on the total number of infections. Our simulations show that, depending on the $\beta^{(i)}$, increasing $p_{travel}$ can decrease or increase the expected total number of infections $\mathbf{E}\left[R_\infty\right]$.

## 1 INTRODUCTION

In this paper we consider the well-known SIR process which is used to study the spread of a contagious disease. The model was introduced in the early 20th century (Ross and Hudson [1917], Kermack and McKendrick [1927]). The population is split into three compartments (or states): susceptible (S), infected (I) and recovered (R). At the beginning of an epidemic, a small number of individuals are infected and the rest of the population is susceptible. Susceptible individuals can be infected and infected individuals can recover and become permanently immune to the disease. The model is often used to study the spread of diseases like COVID-19, measles, mumps and rubella.

When an epidemic starts, the number of infected individuals increases rapidly. At some point, the proportion of still susceptible and already recovered individuals will be such that the spread of the infection will begin to slow down to the point of complete stop. The model is characterised by transition rates between the (compartments/states) depending on the infection rate $\beta$ and the recovery rate $\gamma$.

The so-called reproduction number of an SIR process is defined as $\mathfrak{R}_0 = \beta/\gamma$. This number is equal to the expected number of secondary cases following the introduction of one infected individual into a fully susceptible population. This reproduction number determines how the process evolves over time and how many individuals will get infected until the infection dies out. In general SIR processes behave as follows. If $\mathfrak{R}_0 < 1$, the disease extinguishes early and the number of total infections follows an exponential distribution. If $\mathfrak{R}_0 > 1$, the number of infected individuals first grows exponentially (in expectation). If the number of infected and recovered individuals reaches a fraction of $1 - 1/\mathfrak{R}_0$ the number of infected individuals decreases, in expectation, until it reaches zero. The fraction $1 - 1/\mathfrak{R}_0$ is also called *herd immunity threshold*.

Looking closer at the case $\mathfrak{R}_0 > 1$, one can see that the total number of infections follows a bi-modal distribution. The first peak represents *early extinction*, meaning the process terminates early (see Section 2 for a precise definition). If this unlikely event does not happen, then the process is likely to reach the herd immunity threshold and a much

*Accepted for the 38th Conference on Uncertainty in Artificial Intelligence* (UAI 2022).

larger number of individuals gets infected.

In this paper we study the SIR process as a population protocol. We assume that the population has a fixed size $N$ and that the individuals are modelled as a finite state machine. The state space of our protocol is simple, each individual being in one of the three states $S$, $I$, or $R$. At the beginning of time $t$, we denote by $S_t$, $I_t$ and $R_t$ the number of individuals susceptible to a disease, infected individuals and recovered individuals, respectively. At the beginning of the process $t = 0$, one individual (or a small subset of them) is infected and all other individuals are susceptible. No individual is recovered yet. The process is now defined as follows. In each step a pair of individuals $i, j$ is chosen uniformly at random and is allowed to change state. If the individual $i$ is currently infected, then it recovers with probability $\gamma$. If $i$ is susceptible and $j$ is infected, then $i$ becomes infected with probability $\beta$.

As mentioned before, it may happen that the disease dies out quickly without ever evolving into a pandemic, i.e., without reaching the herd immunity threshold. In this paper we calculate the probability of early extinction as a function of $\mathfrak{R}_0 = \beta/\gamma$ and we present a fully rigorous analysis for such probability using coupling arguments. We show that, if we start with $s$ infected individuals, the probability for early extinction $\mathbf{Pr}\left[\mathcal{E}_{ext}\right]$ is asymptotically $(1/\mathfrak{R}_0)^s$ assuming $\beta > \gamma + \varepsilon$ for some arbitrarily small constant $\varepsilon$. If $\beta < \gamma - \varepsilon$, then $\mathbf{Pr}\left[\mathcal{E}_{ext}\right] = 1 - o(1)$. We then calculate the expected size of the population that gets infected throughout the process $\mathbf{E}\left[R_\infty\right]$. Note that $R_\infty$ denotes the number of recovered individuals at the end of the process. The recovered state is the only terminal state and equals the number of individuals that were infected throughout the process. We first show that $\mathbf{E}\left[R_\infty \mid \mathcal{E}_{ext}\right]$ is close to zero. Then we show, conditioning on $\overline{\mathcal{E}_{ext}}$, that $\mathbf{E}\left[r_\infty \mid \overline{\mathcal{E}_{ext}}\right] = \left(1 + W\left(-\mathfrak{R}_0 \cdot e^{-\mathfrak{R}_0}\right)/\mathfrak{R}_0\right) \pm o(1)$, where $r_\infty = R_\infty/N$. Combining our results, we show that, without any conditioning,

$$\mathbf{E}\left[r_\infty\right] = \mathbf{E}\left[r_\infty \mid \overline{\mathcal{E}_{ext}}\right] \cdot \left(1 - \mathbf{Pr}\left[\mathcal{E}_{ext}\right]\right) \pm o(1).$$

Many of the results found in the literature are based on first-order methods (mean field approaches) (e.g., Kröger and Schlickeiser [2020], Bicher and Popper [2013]) and they give an expression similar to our term for $\mathbf{E}\left[r_\infty \mid \overline{\mathcal{E}_{ext}}\right]$ as an estimation of $\mathbf{E}\left[r_\infty\right]$. This is due to the fact that, with a mean field approach, the process is regarded as deterministic, which means that early extinctions cannot happen for $\mathfrak{R}_0 > 1$. Furthermore, such approach neglects the variance of the process, possibly crucial in the analysis of random processes (see Berenbrink et al. [2017] for a majority type process where the expected change is the same, but the variance in the process determines the convergence time). Although, it has been observed through simulations that the deterministic and stochastic processes differ in terms of the total number of infections (e.g., Fig.13 of Allen and Burgin [2000]).

In the second part of our paper we consider a two-country setting: we show through simulations how travelling affects the spread of the disease. A possible ban, or restriction, on travelling has been considered and applied by governments in some cases, especially with respect to the current pandemic. Such measure falls under the label of non-pharmaceutical interventions (NPI). We introduce an extension of the model where each individual resides in one of two countries. We assume that both countries can have different infection rates, $\beta^{(1)} \neq \beta^{(2)}$, and each country has population size $N^{(1)}$, $N^{(2)}$ with $N = N^{(1)} + N^{(2)}$. The process is similar to our original process. First, an individual $i$ is chosen uniformly at random among the whole population of both countries (with probability $1/N$). The chosen individual then travels to the other country with a probability of $p_{travel}$ and remains in the current country w.p. $1 - p_{travel}$. After this, one of the two countries is chosen with a probability proportional to its population and one step of the single-country process is performed in that country. We study, via simulations, the connection between $p_{travel}$ and the total number of infections over time. We obtain interesting and surprising insights. For example, one would think that increasing $p_{travel}$ increases the total number of infections. While this is generally true for small values of $p_{travel}$, (though for some values of $\beta$ and $\gamma$, not even this is true) it turns out that this does not hold if one country has $\mathfrak{R}_0^{(1)} = \beta^{(1)}/\gamma > 1$ and the other country has sufficiently small $\mathfrak{R}_0^{(2)} < 1$. In that case for large enough values of $p_{travel}$, the total number of infections is decreased. Our simulations suggest that early extinction affects the size of the epidemic, leading its average to be reduced and its variance to increase.

## 1.1 RELATED WORK

The SIR model is a very basic mathematical model for the spread of diseases (e.g. it does not model birth and death of individuals, the latter resulting from the illness or other reasons) and a very large number of generalizations were suggested in the literature. Due to the vast literature, we will only discuss results in the SIR comparable to our results and we will concentrate on analytical results. There are also many compartmental models similar to the SIR model. See Li [2018], Daley [1999] for a nice overview about different mathematical models.

The SIR model was introduced in 1927 by Kermack and McKendrick [1927], building on work by Ross and Hudson [1917]. The authors also introduced the well-known set of non-linear ordinary differential equations (which we introduce in this paper too) to describe the spread of the disease. The authors showed the so-called threshold theorem, which predicts the critical fraction of susceptible individuals in the population that must be exceeded if an epidemic is to occur. The authors show that $s_\infty = 1 - r_\infty =$

$s_0 e^{-\mathfrak{R}_0(r_\infty - r_0)}$, where $s_\infty = |S_\infty|/N$ (fraction of susceptible individuals as time goes to infinity), $r_\infty = |R_\infty|/N$, $s_t = |S_t|/N$, and $r_t = |R_t|/N$. This transcendental equation has a solution in terms of the Lambert $W$ function: $s_\infty = -\mathfrak{R}_0^{-1} W(-s_0 \mathfrak{R}_0 e^{-\mathfrak{R}_0(1-r_0)})$.

Most of the publications investigating the SIR model use numerical methods, employing a wide number of different techniques (see Biazar [2006], Rafei et al. [2007a,b] and the references therein). There is a huge amount of literature studying very diverse effects in the SIR compartmental model. Much of it approximates the random process with a deterministic process defined via ordinary differential equations. Hence an early extinction of the process is not possible. In Harko et al. [2014] the authors derived an exact analytical solution to the SIR model in parametric form. A similar result was shown by Miller [2012, 2017]. In Shabbir et al. [2011] an exact analytical solution to the SIR and SIS models with constant population is obtained with the help of direct integration tools. Lefèvre and Simon [2020] propose a block-structured Markov process to describe the spread of epidemics of SIR type and they determine the distribution of the final state of the process. In Black and Ross [2015] the authors present a new method for the recursive computation of the epidemic size distributions. The authors do not estimate the expected size of the epidemic nor do they give a closed form of its distribution.

Pastor-Satorras et al. [2015] give a great overview about epidemic processes in networks: Here the individuals are connected by a network modelling social contacts, such that the infection spreads from a node to its neighbours. The authors of Janson et al. [2014] consider random graphs with a given degree sequence and prove that there is a threshold as a function for $\gamma$, $\beta$, and given vertex degrees. Below the threshold, only a small number of infections occurs, while above it most of the graph gets infected. Kempe et al. [2003] consider influence maximisation in the independent cascade model introduced in Goldenberg et al. [2001a,b]. The process works in parallel rounds and it starts with a set of active (infected) nodes. Every active node infects every non-infected neighbour with a probability of $p$. Then the active nodes become inactive and the newly infected neighbours become active. The process runs until no more activations are possible. The optimization problem of selecting the most influential nodes to be activated in the beginning is NP-hard and the authors provide the first provable approximation guarantees for efficient algorithms. For an overview about results in this model see Shakarian et al. [2015].

Bohman and Picollelli [2012] consider an SIR process of random graphs with a given degree sequence in an continuous time model. Each infected node sends infections to each of its neighbors at times determined by independent exponential random variables with parameter $\lambda$. An infected node recovers at a random time given by an independent exponential random variable with parameter $r_\infty$. The authors assume that the infection spreads from a single infected node and show that either the disease halts after infecting only a small number of nodes, or an epidemic spreads to infect a linear number of nodes. The authors also show that, conditioned on the event that more than a small number of nodes are infected, the epidemic is likely to follow a trajectory given by the solution of an associated system of ordinary differential equations. Their approach gives bounds on the total number of infected nodes.

There is also related work that considers an agent-based modelling of the SIR model (e.g. Bicher and Popper [2013]). These investigate some form of non-homogeneous populations (e.g. distances between agents given by a graph), and analyse results empirically, either using geographic information systems Perez and Dragicevic [2009], or social relationships Alzu'bi et al. [2021].

Many works model how different societal factors play a role in the evolution of an epidemic. In the case of the COVID-19 pandemic, many of them focus on accounting for different factors, like social distancing measures and testing regimens in order to build a reliable model. In Levesque et al. [2021] the authors create a Crump-Mode-Jagers continuous time branching process modelling COVID-19 propagation in order to decide which mitigation strategies are more effective. Similar works developed mathematical and data-driven models in order to establish the efficacy of such measures Sun et al. [2020], Choi and Shim [2021], Liu et al. [2021]. However, there is still a lack of work studying the role travelling has on the epidemic. In Arino and Van Den Driessche [2003], the authors incorporate the concept of travel into the ordinary differential equations for a Susceptible-Exposed-Infected-Removed-Susceptible (SEIRS) model. They derive bounds on $\mathfrak{R}_0$ and show that a disease-free equilibrium is reached, though they hint at its uniqueness through simulations and no discussion is made about the final expected size of the population. Their model is complex and detailed, with the central assumption that individuals come back to the origin country before leaving again for any other country. This is different from our work, where we do not require individuals to follow travel patterns. In Zakary et al. [2017] a multi-regions discrete-time model describes the spatial-temporal spread of an epidemic. Starting from one region, this enters to regions connected with their neighbors by any mean favouring movement. Like in our case, the authors consider homogeneous SIR populations.

## 2 MODELS AND RESULTS

In this section we introduce formally the two models considered in this paper. Both models work in sequential rounds and are population protocols Angluin et al. [2006]. At every round, two individuals $i$ and $j$ are picked uniformly at random. Individual $i$ can either become infected, recover, or stay susceptible, depending on the current state of $i$ and $j$. In

our or second model, individuals can also travel. We study discrete-time models, where in each integer step, an action occurs. One major advantage the discrete-time models have over continuous time models is that they allow to study the early extinction phenomenon. Note that continuous systems and, in particular, differential equations fail to capture early extinctions.

**Single Country Model.** Let $S_t, I_t, R_t$ be the number of susceptible, infected and recovered individuals at time $t$, with $S_t + I_t + R_t = N$, where $N$ is the total number of individuals. Our SIR process is defined as follows. In every round two individuals $i, j$ are picked uniformly at random.

1. If individual $i$ is infected, then it recovers w.p. $\gamma$.
2. If individual $i$ is susceptible and $j$ is infected, then $i$ will becomes infected w.p. $\beta$.

Note that $\beta \cdot I_t / N$ is the probability that the selected susceptible individual $i$ becomes infected. We assume that $\beta$ and $\gamma$ are both constants. In expectation, the system evolves as follows.

$$\mathbf{E}\left[I_{t+1}\right] = I_t + \beta(I_t/N)(S_t/N) - \gamma(I_t/N) \quad (1)$$
$$\mathbf{E}\left[S_{t+1}\right] = S_t - \beta(I_t/N)(S_t/N) \quad (2)$$
$$\mathbf{E}\left[R_{t+1}\right] = R_t + \gamma(I_t/N) \quad (3)$$

The reproduction number is defined as $\mathfrak{R}_0 = \beta/\gamma$. This number is equal to the expected number of infections caused by an infected individual assuming that $S = N$. Over time, the number $S_t$ decreases such that every newly infected individual introduces less and less infections into the population. The herd immunity threshold is defined as $1 - 1/\mathfrak{R}_0$. This is the value of $S_t/N$ such that $\mathfrak{R}_0 \cdot S_t/N = 1$.

In Last [2001], a pandemic is defined as "an epidemic occurring worldwide, or over a very wide area, crossing international boundaries and usually affecting a large number of individuals". In this work we assume a simplified notion of pandemic, depending only on the total number of infected individuals. We say a process results in a *pandemic* if at one point of time the number of active infections ($I_t$) reaches $\sqrt{N}$.[1] Moreover, if the process does not result in a pandemic, then we say it *extincts early*. We denote the corresponding events by $\mathcal{E}_{pan}$ and $\mathcal{E}_{ext}$. Theoretically, over time a constant fraction of the population might become infected, even though at any point of time the number of current infections stays below $\sqrt{N}$. However, with the same techniques as we use in this paper, one can show that this

---

[1]Our results suggest that any value $> \log N$ scaled with $\beta$ and $\gamma$ will work, as reaching it will ensure that eventually a constant fraction of $N$ will become infected with very high probability. The reason for $\sqrt{N}$ is it avoids using terms of $\gamma$ and $\beta$ in the definition, assuming they are both constants (independent of $N$). There is virtually no difference between a threshold of say $C \log n$ and any value in $n^{1-\varepsilon}$ for constants $C = C(\beta, \gamma)$ and $\varepsilon < 1$, since, if an infection reaches size $C \log n$, then it will also reach $n^{1-\varepsilon}$ with high probability.

event happens with an exponentially small probability—it is akin to a biased random walk on a line remaining in the interval $(1, \sqrt{N})$ for a linear number of rounds. Finally, we define $R_\infty$ as the *epidemic* final size; since all individuals eventually recover this corresponds to $R_\infty = \lim_{t \to \infty} R_t$. We define $r_\infty = R_\infty/N$. Let $W$ denote the Lambert W-function, which is the inverse of function $f(W) = We^W$ considered for $W \in [-1, \infty)$ (where function $f$ increases monotonically from $-1/e$ to infinity).

**Model for two countries.** In this case we assume that the $N$ individuals are distributed over two countries. $N_t^{(1)}$ and $N_t^{(2)}$ denotes the number of individuals staying in country 1 and 2 at time $t$. We assume $N_t^{(1)} = N_t^{(1)} = N/2$. The number of susceptible, infected and recovered individuals at time $t$ in country $i$ is denoted by $S_t^{(i)}, I_t^{(i)}$ and $R_t^{(1i)}$ for $i \in \{1, 2\}$. Initially we have $I_t^{(1)} = I_t^{(2)} = s$ for $s \geq 1$. We assume that every country has its own infection rate: $\beta^{(1)}$ and $\beta^{(2)}$. The recovery rate in both countries is $\gamma$. The process has two selection steps and step $t$ works as follows.

1. Pick an individual $i$ uniformly at random (with prob. $1/N$). With probability $p_{travel}$ individual $i$ travels from its current country to the other country. Adjust $N_t^{(1)}$ and $N_t^{(2)}$ accordingly.

2. Pick country $\ell$ as follows: $\ell = 1$ w.p. $N_t^{(1)}/N$ and 2 otherwise.
   An SIR step as described above is applied on the chosen country.
   (a) Pick a pair of individuals $i$ and $j$ uniformly at random from country $\ell$.
   (b) If individual $i$ is infected, then it recovers w.p. $\gamma$.
   (c) If individual $i$ is susceptible and $j$ is infected, then $i$ will become infected w.p. $\beta^{(\ell)}$.

Compared to existing works, this model is simpler and posing minimal constraints to the movement of individuals. This makes it easily adaptable, for example in the case of a change in the number of countries involved. Further, a simpler model might facilitate future analytical results regarding the expected size of the epidemic.

## 2.1 RESULTS

Our main result Theorem 4 shows a very tight characterisation of the early extinction probability. Recall that a process extincts early if the number of infected dies out before ever reaching $I_t \geq \sqrt{N}$, where $N$ is the size of the population. Recall that $\mathfrak{R}_0 = \beta/\gamma$ is the reproduction number, $R_\infty$ is the total number of infected individuals and $r_\infty = R_\infty/N$. First we present a simplified version of Theorem 4.

**Theorem 1** (Simplified version of Theorem 4)**.** *Let $\beta + \gamma \leq 1$. Consider the single country model starting with*

$I_0 = s \leq \frac{\sqrt{N}}{2 \log N \log \log N}$ *infections. Let* $\varepsilon = 5 \log N / \sqrt{N}$ *and let* $\mathbf{Pr}\left[\mathcal{E}_{ext}\right]$ *be the probability of early extinction.*

1. *If* $\beta < \gamma - \varepsilon$*, then* $\mathbf{Pr}\left[\mathcal{E}_{ext}\right] \geq 1 - o(1)$.

2. *If* $\beta > \gamma + \varepsilon$*, then* $\mathbf{Pr}\left[\mathcal{E}_{ext}\right] = (1/\mathfrak{R}_0)^s \pm o(1)$.

For the case where $\beta > \gamma$ and $s = 1$ the probability of early extinction is essentially $1/\mathfrak{R}_0 = \gamma/\beta$. If $s > 1$ the probability decreases exponentially in $s$. If $\beta < \gamma$, then the process is very likely to reach an early extinction. We leave the case $\beta = \gamma$ open but note that $\varepsilon$ will be arbitrary small with growing $N$. The challenging part in the proof of Theorem 4 is non-linear update rule of our process. Note that rewriting (1)-(3) by replacing $R_t = N - S_t - I_t$ reveals the non-linearity. To overcome these challenges, we use a series of couplings, allowing us to relate our SIR process to a biased random walk.

Note that in the above result $\mathcal{E}_{ext}$ is defined as the event of having $\sqrt{N}$ many infected individuals at the same time. Hence, it is still possible that $R_\infty = \Omega(N)$, i.e., at the end of the process the number of recovered individuals (which equals the total number of infections) is linear in $N$. In Proposition 1 (see Section 3.1) we show that this is not the case; in the event of $\mathcal{E}_{ext}$ the number of recovered individuals never exceeds $\sqrt{N} \log N$, w.h.p. This means that our bounds on the the early extinction still hold even if the definition was changed to requiring $R_\infty \leq \sqrt{N} \log N$.

The next theorem shows a bound on the the expected number of total infections $\mathbf{E}\left[R_\infty\right]$, expressed in terms of the Lambert function $W$. Note that $W(x) < 0$ for $x \in [-1/e, 0)$. As mentioned before, many approaches like first-order methods, mean field approaches and ordinary differential equations ODE (see e.g., Kröger and Schlickeiser [2020], Bicher and Popper [2013]), cannot account for early extinction due to the underlying determinism. Instead, they obtain bounds of the kind $\mathbf{E}\left[r_\infty \mid \overline{\mathcal{E}_{ext}}\right]$. There has also been some work based on stochastic differential equations (e.g., Allen [2008], Williams et al. [2012]), complemented with simulations results. We are not aware of results based on SDEs that obtain closed-form results for $\mathbf{E}\left[r_\infty\right]$.

**Theorem 2.** *Assume that* $|\beta - \gamma| \geq \varepsilon$ *for an arbitrarily small constant* $\varepsilon$*. Assume* $N = N(\varepsilon)$ *is large enough. Let* $W : [-1/e, \infty) \longrightarrow [-1, \infty)$ *be the Lambert function. Consider the single country model starting with* $I_0 = s \leq \frac{\sqrt{N}}{2 \log N \log \log N}$ *infections.*

1. *If* $\beta < \gamma - \varepsilon$*, then the expected total number of infections is sublinear, i.e.,* $\mathbf{E}\left[r_\infty\right] = o(1)$.

2. *If* $\beta > \gamma + \varepsilon$*, then*
$$\mathbf{E}\left[r_\infty\right] \sim \left(1 + \frac{W\left(-\mathfrak{R}_0 \cdot e^{-\mathfrak{R}_0}\right)}{\mathfrak{R}_0}\right)\left(1 - \left(\frac{1}{\mathfrak{R}_0}\right)^s\right).$$

*See Fig. 1 for a depiction.*

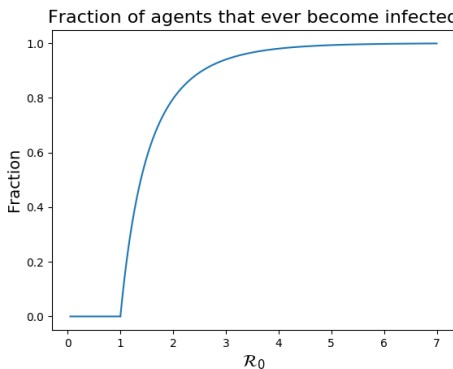

Fraction of agents that ever become infected

Figure 1: The Plot Shows $r_\infty$ for Different Values of $\mathfrak{R}_0 = \beta/\gamma$ and $s = 1$.

The following theorem estimates the expected number of infections conditioned on not having early extinction. It is used to show Theorem 2. The following theorem yields a result similar to the one obtained through first-order methods used by Allen [2008], Williams et al. [2012], but adapted to our population based model. In Theorem 6 (in Section 4) we provide a generalisation of the following theorem.

**Theorem 3.** *Consider the single country model. Assume that* $\beta > \gamma$ *are constants. Let* $W : [-1/e, \infty) \longrightarrow [-1, \infty)$ *be the Lambert function. Then, for large* $n$,

$$\mathbf{E}\left[r_\infty \mid \overline{\mathcal{E}_{ext}}\right] \sim \left(1 + \frac{W\left(-\mathfrak{R}_0 \cdot e^{-\mathfrak{R}_0}\right)}{\mathfrak{R}_0}\right). \quad (4)$$

*Furthermore,* $r_\infty$ *is concentrated around its expected value.*

## 3 EARLY EXTINCTION

The SIR model described in Section 2, often results in herd-immunity, where a large fraction of the population was infected such that only very few susceptible individuals remain. From then on, the number of infections decreases slowly until it finally reaches zero. However, in some case, even when $\beta > \gamma$, it can happen that the number of infections remains low and the virus vanishes very suddenly. This is what we refer to as an *early extinction* and which is the focal point of this section. We will derive bounds on the probability of an early extinction in terms of the parameters $\gamma$ and $\beta$. The formal statement is given in the next theorem and implies the theorem given in Theorem 1.

**Theorem 4.** *Consider the single country SIR model as described in Section 2. Define* $\tau = \sqrt{N} \log N$, $\varepsilon = 5 \log N / \sqrt{N}$, *and* $\varepsilon' = \varepsilon(N - \tau)/N$, *and assume* $I_0 = s$ *for any* $s \leq \frac{\sqrt{N}}{2 \log N \log \log N}$. *Then we have for* $\beta > \gamma + \varepsilon$

$$\left(\frac{\gamma}{\beta}\right)^s - \frac{3}{N} \leq \mathbf{Pr}\left[\mathcal{E}_{ext}\right] \leq \left(1 + \frac{s\tau}{N - s\tau}\right)\left(\frac{\gamma}{\beta}\right)^s + \frac{2}{N}$$

*and for* $\beta < \gamma - \varepsilon$

$$1 - \left(\frac{\gamma}{\beta}\right)^{s-\sqrt{N}} - \frac{2}{N} \le \mathbf{Pr}\left[\mathcal{E}_{ext}\right] \le 1$$

To show Theorem 4 our approach is as follows. Ideally, we would like to bound the number of infections as a (biased) random walk over the number of infected individuals. The probability that such a random walk reaches either of its end points are well-understood. Unfortunately, here the probability to 'walk' from $I_t$ to $I_t - 1$ or $I_t + 1$ is a function of both $I_t$ and $S_t$. We avoid the dependency on $I_t$ by considering *active* steps only. Recall, these are steps in which the number of infected individuals either increases by one or decreases by one. This allows us to drop the terms $I_t/N$ in the transition probabilities. We circumvent the dependency on $S_t$ by defining a family of processes and by coupling them with which each other (see Section 3.1). The processes will be defined and motivated later in this section. Once both problems are solved, we are left with four biased random walks (upper vs lower bounds and $\beta < \gamma$ vs $\beta > \gamma$). This allows us in Theorem 4 to derive almost tight bounds on $\mathbf{Pr}\left[\mathcal{E}_{ext}\right]$.

## 3.1 COUPLINGS

In this section our goal is to show Theorem 5 which upper and lower bounds the probability of early extinction of our process by the extinction probability of two biased random walk process. Theorem 5 shows that the approximation error is tiny and has only a second-order impact on the extinction probability. In order to prove Theorem 5, we will define two intermediate processes and provide three pairwise couplings.

In this section, unless stated otherwise, we only consider active steps, i.e., steps where $I_{t+1} \ne I_t$. The subscript $t$ counts now only the active steps. Recall that $\mathcal{E}_{ext}$ (early extinction) is the event that the number of infections reaches zero before ever having $\sqrt{N}$ infected individuals at the same time. Define stopping time as $T = \min_t \left\{I_t = 0 \text{ or } I_t = \sqrt{N}\right\}$. Hence, $T$ refers to the number of active steps until there are either zero or $\sqrt{N}$ many infected individuals, and $\mathcal{E}_{ext}$ is the event that $I_T = 0$. For any SIR process $P$, we denote by $\xi_t^{(P)}$ the configuration of $P$ at time $t$, which is given by the triplet $\xi_t^{(P)} = \left(S_t^{(P)}, I_t^{(P)}, R_t^{(P)}\right)$. We define $\tau = \sqrt{N} \log N$ and a second stopping time $T_\tau = \min_t\{I_t = 0, I_t = \sqrt{N} \text{ or } t = \tau\} = \min\{T, \tau\}$.

Compared to the stopping time $T$, here the process does not only stop if the number of infected individuals reaches zero or $\sqrt{N}$ but also if that does not happen during the first $\tau$ (active) time steps. All our processes are stopped at time either $T$ or $T_\tau$. We assume that for every $t$ larger that the stopping time we have $\xi_t^{(P)} = \xi_T^{(P)}$ or $\xi_t^{(P)} = \xi_{T_\tau}^{(P)}$, respectively. Now we are ready to define our processes, which we call $P, P_{\tau,*}, P_{*,S}, P_{\tau,S}$.

| | no time limit | time limit $\tau$ |
|---|---|---|
| $S$ not fixed | $P$ | $P_{\tau,*}$ |
| $S$ fixed | $P_{*,S}$ | $P_{\tau,S}$ |

Table 1: Description of the processes.

$$P \xrightarrow{\text{Prop 1}} P_{\tau,*} \xleftarrow{\text{Prop 2}} P_{*,S} \xleftarrow{\text{Prop 3}} P_{\tau,S}$$

Figure 2: Diagram of the couplings

1. $P$ is our original process with the stopping time $T$.
2. $P_{\tau,*}$ is our original process with the stopping time $T_\tau$.
3. Fixing $S_t$ over time to a constant $S \in \{N - \tau, N\}$, we define the following biased random walk process $P_{*,S}$. At each round the following is done.
   (a) Draw random numbers $X_I, X_S, X_a$ i.i.d. and u.a.r. from $[0, 1]$
   (b) If $X_I \in [0, I_t^{(P_{*,S})}/N)$, then
       i. if $X_a \in [0, \beta)$ and $X_s \in [0, S/N]$, increase $I_t^{(P_{*,S})}$;
       ii. if $X_a \in [\beta, \beta + \gamma)$, decrease $I_t^{(P_{*,S})}$.
   The nice property about process $P_{*,S}$ is that (since we only consider active time steps) the process behaves exactly like a biased random walk. The increase probability of $I_t$ is $\beta S/N$ and the decreas probability is $\gamma$. For this process we apply the stopping time $T$.
4. For $S \in \{N - \tau, N\}$, we define the process $P_{\tau,S}$. $P_{\tau,S}$ behaves like $P_{*,S}$ but we apply the stopping time $T_\tau$.

Table 1 sums up the four processes, while Fig. 2 shows the couplings we are going to prove in the following sections. In this section we will show the following result.

**Theorem 5.** *Consider the processes $P$ and $P_{*,S}$. We have*

$$\mathbf{Pr}\left[I_T^{(P_*,N-\tau)} = \sqrt{N}\right] - 2/N \le \mathbf{Pr}\left[I_T^{(P)} = \sqrt{N}\right] \le$$
$$\le \mathbf{Pr}\left[I_T^{(P_*,N)} = \sqrt{N}\right] + 2/N, \quad and$$

$$\mathbf{Pr}\left[I_T^{(P_*,N)} = 0\right] - 2/N \le$$
$$\mathbf{Pr}\left[I_T^{(P)} = 0\right] \le \mathbf{Pr}\left[I_T^{(P_*,N-\tau)} = 0\right] + 2/N$$

From this it follows that we can analyse the early extinction time of $P_{*,S}$ instead of $P$. To show Theorem 5 we use 3 couplings: Proposition 1 couples processes $P$ and $P_{\tau,*}$, Proposition 2 couples $P_{\tau,*}$ and $P_{\tau,S}$, and Proposition 3 connects $P_{\tau,S}$ and $P_{*,S}$.

**Proposition 1.** *With prob $1 - 1/N$ we have $I_T^{(P)} = I_{T_\tau}^{(P_{\tau,*})}$.*

**Proposition 2.** *Consider the processes $P_{\tau,*}$ and $P_{\tau,S}$ for $S \in \{N, N - \tau\}$. Let $P = \mathbf{Pr}\left[I_{T_\tau}^{(P_{\tau,*})} = \sqrt{N}\right]$. We have*
$$\mathbf{Pr}\left[I_{T_\tau}^{(P_\tau,N-\tau)} = \sqrt{N}\right] \le P \le \mathbf{Pr}\left[I_{T_\tau}^{(P_\tau,N)} = \sqrt{N}\right].$$

**Proposition 3.** *Consider the processes $P_{\tau,S}$ and $P_{*,S}$ for $S \in \{N - \tau, N\}$. Then we have*

$$\mathbf{Pr}\left[I_{T_\tau}^{(P_{\tau,N})} = \sqrt{N}\right] \leq \mathbf{Pr}\left[I_T^{(P_{*,N})} = \sqrt{N}\right] + 1/N, \ and$$

$$\mathbf{Pr}\left[I_T^{(P_{*,N-\tau})} = \sqrt{N}\right] - 1/N \leq \mathbf{Pr}\left[I_{T_\tau}^{(P_{\tau,N-\tau})} = \sqrt{N}\right]$$

*Proof.* This proof is similar to proof of Proposition 1. □

Putting all three couplings together yields Theorem 5. From this we are finally able to conclude Theorem 4, since now we can analyse a biased random walk instead. The proof distinguishes between two cases, depending on whether $\beta$ or $\gamma$ is larger and builds on known results for biased random walks. It can be found in the supplementary material.

## 4 TOTAL NUMBER OF INFECTIONS

In this section we analyse the total number of infections for the SIR model for one country. The proof can be found in the full version.

**Theorem 6.** *Consider the single country model. Assume that $\beta - \gamma \geq \varepsilon$ for an arbitrarily small constant $\varepsilon$. Assume $N = N(\varepsilon)$ is large enough. Let $W : [-1/e, \infty) \longrightarrow [-1, \infty)$ be the Lambert function. Consider a time step $t^* = \omega(1)$. Let $\mathcal{E}'$ be the event $I_{t^*} \in [\omega(1), o(N)]$ and $R_{t^*} = o(N)$. Then, for large $n$,*

$$\mathbf{E}\left[r_\infty \mid \mathcal{E}'\right] \sim \left(1 + \frac{W\left(-\mathfrak{R}_0 \cdot e^{-\mathfrak{R}_0}\right)}{\mathfrak{R}_0}\right). \qquad (5)$$

*Furthermore, $r_\infty$ is concentrated around its expected value. Moreover, $\mathbf{E}\left[r_\infty \mid \mathcal{E}'\right]$ is concave for $\mathfrak{R}_0$.*

## 5 SIMULATIONS FOR TWO COUNTRIES

In this section, we study the impact of travelling and its relationship to early extinction on the total number of infections. Recall that $p_{travel}$ is defined as the travelling rate. Without loss of generality we set $\gamma = 0, 2$ in all our simulations.[2] We set the initial size of each country to be $N_0^{(1)} = N_0^{(2)} = 2000$. Note that these size vary to a small extent throughout the process due to traveling individuals. The total population is $N = N_0^{(1)} + N_0^{(2)} = 4000$. Initially, there is one infection per country ($I_0^{(1)} = I_0^{(2)} = 1$) and the rest of the population is susceptible. In our simulations we vary the travelling rate $p_{travel}$ and the infection rates $\beta^{(1)}$ and $\beta^{(2)}$. W.l.o.g. we assume $\beta^{(1)} \leq \beta^{(2)}$. For each value of $p_{travel}$, $\beta^{(1)}$, and $\beta^{(2)}$, we output the average of

---

[2]By changing $\beta$ one can get arbitrary reproduction numbers $\mathfrak{R}_0$. Similarly, one can vary $p_{travel}$ to obtain arbitrary ratios of $p_{travel}/\gamma$.

10.000 iterations. The outcome of the simulations can be found in Fig. 3 and Fig. 4. For each value of $p_{travel}$ we plot the average number of infections $\widehat{R_\infty^{(1)}}$ in country one in light green and in light blue we plot the average number of infections in country two $\widehat{R_\infty^{(2)}}$. Note that we write $\widehat{R_\infty}$ instead of $\mathbf{E}\left[R_\infty\right]$ to emphasise the difference between the average and the expected value. The total number of infections across both countries $\widehat{R_\infty} = \widehat{R_\infty^{(1)}} + \widehat{R_\infty^{(2)}}$ is plotted in red. We plot the smoothed version of $\widehat{R_\infty}$ in black. Let $R_\infty(p_{travel})$ denote the total number of infections for a given travelling rate $p_{travel}$. Finally, the corresponding standard deviation is plotted using vertical bars.

First, note that the standard deviation appears large: this is inherent to the process and due to the fact that early extinctions (in this case, when $\widehat{R_\infty} \leq \sqrt{2N} = \sqrt{4000} \approx 63$) occur with constant probability. Indeed, the plots are the result of averaging among all the iterations, included the ones resolving in early extinction. By excluding runs with an early extinction, the standard deviation becomes negligible. Another effect of this exclusion is that the values of $\widehat{R_\infty}$ turn up to be visibly smaller (see Fig. 3). Another interesting detail with respect to this is shown in Fig. 4a. In there, the error bars representing the variance significantly decrease with increasing $p_{travel}$. This tells us that $\widehat{R_\infty}$ becomes small because the percentage of runs that terminate with early extinction increases significantly, compared to the case where no or little travel is present. This to the point that these runs do not constitute the variance of the process anymore, but the most likely outcome, hence the reduction in the variance.

It is perhaps natural to assume that increasing $p_{travel}$ also increases $\widehat{R_\infty}$. While this is true for some values of the $\beta$s, there are also values for which $\widehat{R_\infty}$ first increases monotonically and then decreases monotonically. In addition, for large $p_{travel}$ the number of total infections can drop below the value where no travelling occurs, i.e., $\widehat{R_\infty}(1) < \widehat{R_\infty}(0)$. Table 2, Fig. 3 and Fig. 4.

**Simulation Result 1.** *Assume that $\gamma = 0.2$.*

1. *$\beta^{(1)} = 0$ and $\beta^{(2)} = 0.3$ results in $R_\infty(p_{travel})$ being non-monotone. Moreover, $R_\infty(1/2) = O(1)$ (Fig. 4a).*

2. *$\beta^{(1)} = 0$ and $\beta^{(2)} = 0.5$ results in $R_\infty(p_{travel})$ being non-monotone. (Fig. 4b).*

3. *$\beta^{(1)} = 0.5$ and $\beta^{(2)} = 0.8$ results in $R_\infty(p_{travel})$ increasing monotonically. (Table 2, and also Fig. 4c).*

Based on this, we believe that the simulation results can be generalised to the following conjecture.

**Conjecture 1.** *For all $\beta^{(1)}, \beta^{(2)}$ and $\gamma$ we have*

*(i) $\beta^{(1)}, \beta^{(2)} < \gamma$: $\mathbf{E}\left[R_\infty(p_{travel})\right] = O(1)$ for all $p_{travel} \in [0, 1]$.*

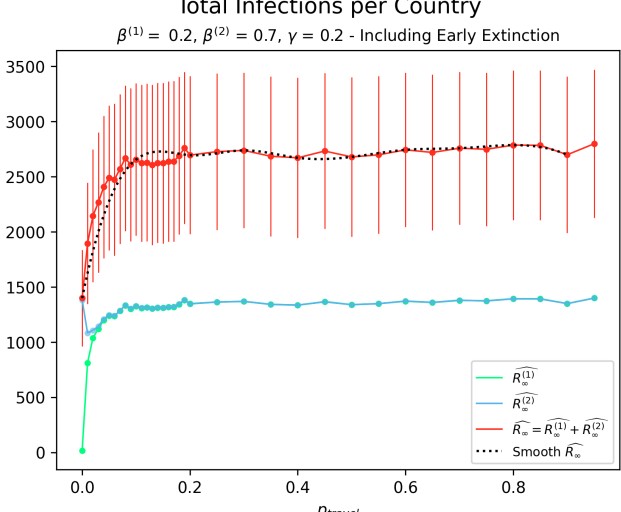

(a) Including Early Extinction.

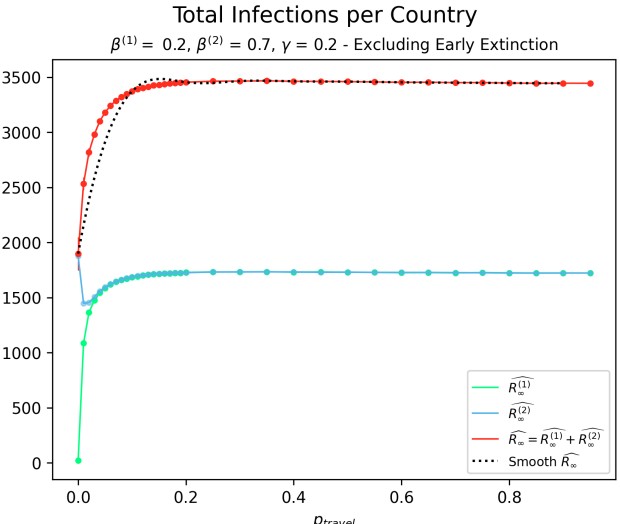

(b) Excluding Early Extinction.

Figure 3: The Plots Show $r_\infty$ as Function of $p_{travel}$ Averaged Across 1000 Iterations. We Consider two Different Settings. In the Plot of the l.h.s. we Plot $\widehat{R_\infty}$. For the Plot on the r.h.s. we Plot $\widehat{R_\infty}$ Conditioned on $\overline{\mathcal{E}_{ext}}$. The Plots Shows that $\widehat{R_\infty}$ Conditioned on $\overline{\mathcal{E}_{ext}}$ has Only very Little Standard Deviation whereas the Standard Deviation of $\widehat{R_\infty}$ is Considerably Larger. However, this is Unavoidable since there is a Positive Probability of an Early Extinction in which Case the Total Infections is Close to 0. Therefore, the Standard Deviation is Naturally of Linear Size. Note that in All Other Experiments we have Ten Times More Iterations Suggesting an Adequate Number of Iterations in our Experiments.

(ii) $\beta^{(1)} < \gamma < \beta^{(2)}$ is non-monotone.

(iii) $\beta^{(1)}, \beta^{(2)} > \gamma$: $\quad \mathbf{E}\left[R_\infty(p_{travel})\right]$ increases monotonically. Note that Fig. 4c suggests that it is important

Table 2: The Table Shows Our Simulation Results for $\widehat{R_\infty}(p_{travel})$ Given Different Values of $\beta$s over 10.000 Iterations, with the Exception of $(0.2, 0.7)$ – Where Early Extinction (EE) Runs are Excluded (*) – that is Averaged Over 1000 Iterations. We Round to the Nearest Integer.

| $\beta^{(1)}, \beta^{(2)}$ \ $p_{travel}$ | 0 | 0.05 | 0.1 | 0.2 | 0.5 | 0.95 |
|---|---|---|---|---|---|---|
| 0, 0.3 | 395 | 293 | 121 | 32 | 12 | 10 |
| 0, 0.5 | 1078 | 1514 | 1491 | 1352 | 1025 | 857 |
| 0, 0.8 | 1470 | 2266 | 2416 | 2497 | 2499 | 2500 |
| 0.2, 0.7 | 1399 | 2489 | 2655 | 2695 | 2679 | 2798 |
| (*) 0.2, 0.7 | 1899 | 3181 | 3373 | 3454 | 3460 | 3445 |
| 0.3, 0.3 | 763 | 1276 | 1301 | 1282 | 1305 | 1294 |
| 0.3, 0.5 | 1467 | 2357 | 2433 | 2464 | 2419 | 2416 |
| 0.3, 0.8 | 1868 | 2926 | 3085 | 3132 | 3189 | 3185 |
| 0.5, 0.8 | 2543 | 3367 | 3407 | 3428 | 3443 | 3465 |
| 0.8, 0.8 | 2953 | 3673 | 3687 | 3688 | 3679 | 3682 |

*for the $\beta$s to be strictly larger than $\gamma$ in order for $\mathbf{E}\left[R_\infty(p_{travel})\right]$ to increase monotonically.*

At the core of the conjecture is the following observation. As $p_{travel}$ increases, both $\beta^{(1)}$ and $\beta^{(2)}$ are blended together resulting in a linear combination of both values. Consider the following thought experiment where the $\beta$s fully blend until the "effective" $\beta$ of both cities is the average $\overline{\beta} = (\beta^{(1)} + \beta^{(2)})/2$. If we ignore the effect of travelling, all statements of the conjecture follow from the Theorem 2. On the other hand, for high values of $p_{travel}$, then we can look at $\overline{\beta}$ and see in Fig. 4a that if $\overline{\beta}$ is below $\gamma$, then the total number of infections $R_\infty$ is close to 0. We give the following explanation. When travel becomes more likely, infected agents might be selected to travel from one country to the other one. In such starting country, then, there will be less infected agents, making the infection step even more less likely than the recovery step. This further reduces the number of infected agents. On top of this, recovered or susceptible agents can be selected to travel too, lowering the ratio of infected agents in their destination country. Since the process starts with a small number of infected agents, travelling will undermine their influence significantly more than the influence of susceptible or recovered agents.

For $\overline{\beta}$ larger than $\gamma$, $R_\infty$ is concave (see Theorem 6). The concavity is the key to the understanding. For example, when both original $\beta$s were above $\gamma$, then having $\overline{\beta}$ in both countries increases $R_\infty$, due to the concavity. Let us consider the cases of the conjecture one by one. Part (i) follows since $\overline{\beta} < \gamma$ and any blending of the $\beta$s will yield "effective" $\beta$s less than $\gamma$. For part (ii) note that when $p_{travel} = 0$ and no blending of the $\beta$s occurs, then we have that in one city $\mathfrak{R}_0 > 1$ (expected pandemic) and in the other $\mathfrak{R}_0 < 1$ (expected early extinction). When $p_{travel}$ is slightly above 0 then the blending of the $\beta$s effectively decreases $\beta^{(2)}$ a little bit, the number of infectable individuals also doubles once $p_{travel} > 0$. This becomes clear when one considers the set-

ting where $\beta^{(1)} = 0$ and $\beta^{(2)} = 1$. Here $\mathbf{E}[R_\infty] > N$ and thus clearly some individuals of country 1 contribute. Part (iii) follows immediately from the concavity. The simulations presented here are only a small fraction of the number of our simulations covering the large variety of different values for $\beta^{(1)}, \beta^{(2)}$ and $\gamma$. All our simulations confirmed our conjecture. Nonetheless, we were not able to turn the arguments above into a rigorous proof. On one hand, considering one country only is already immensely complicated and a second country increases the number of variables and their decencies even further. On the other hand, $\mathbf{Pr}[\mathcal{E}_{ext}]$ and further considerations have to be accounted for. Indeed, one cannot simply ignore $p_{travel}$ and assume that both countries have $\overline{\beta}$. The simulations suggest that we have essentially two values for $R_\infty$; one being $R_\infty(0)$ and one being $R_\infty(p)$, for any $p > 0$. The reason for this is that if $p_{travel} = 0$, then it can happen that one city has an early extinction. When $p_{travel} > 0$, we believe that whenever only one of the countries has an early extinction, then the other country will eventually send infected individuals over and rekindle the infection until it succeeds. While the above shows the influence of effects when $p_{travel}$ is small, ww believe that for large values of $p_{travel}$ our arguments capture the behaviour of the process.

# 6 FUTURE WORK

In this work we adopted one of the many standard models for disease spreading assuming pairwise interactions. Incorporating super spreaders is a great idea for further work. One way to incorporate super spreaders is to assume an underlying social network with high degree nodes. Another one is to first activate a random individual and then to choose a random number of interaction partners using a suitable distribution. Both models need a different analysis method compared to the analysis in this paper. We believe it is very interesting to study them in future work.

**Acknowledgements**

F. Mallmann-Trenn was in part supported by the EPSRC grant EP/W005573/1.

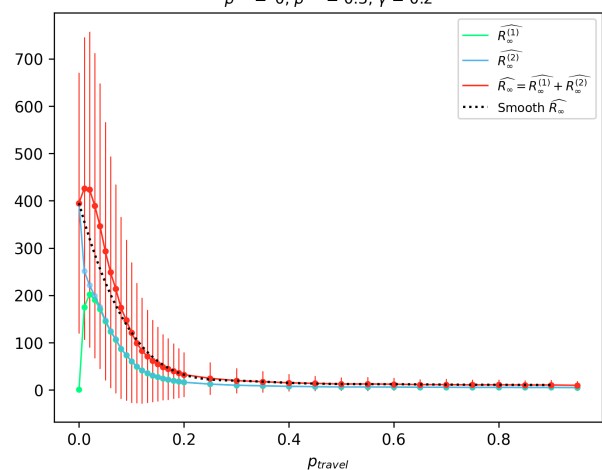

(a) $\beta^{(1)} < \gamma < \beta^{(2)}$ and $\overline{\beta} < \gamma$. The Total Number of Infections $\widehat{R_\infty}(p_{travel})$ Appears to be non-Monotone.

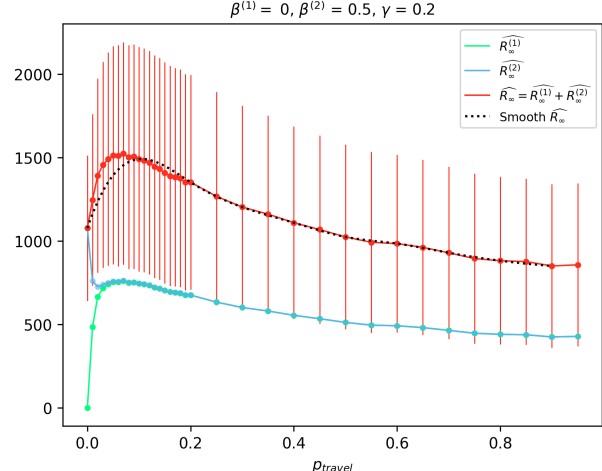

(b) $\beta^{(1)} < \gamma < \beta^{(2)}$ and $\overline{\beta} > \gamma$. The Total Number of Infections $\widehat{R_\infty}(p_{travel})$ Appears to be non-Monotone.

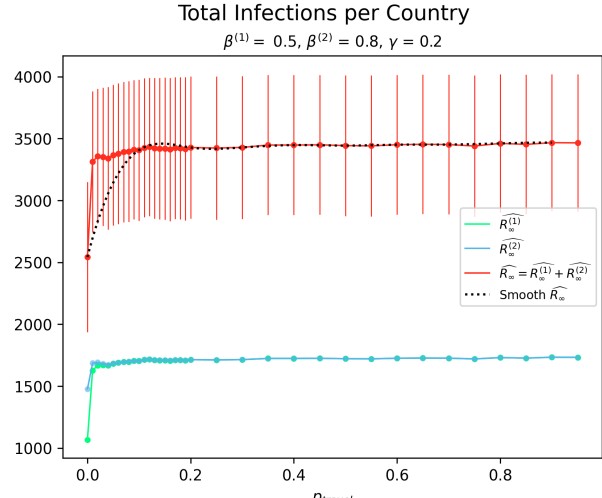

(c) $\beta^{(1)}, \beta^{(2)} > \gamma$. The Total Number of Infections $\widehat{R_\infty}(p_{travel})$ Appears to be Monotonically Increasing. See also Table 2.

Figure 4: Simulation results for $\widehat{R_\infty}$ varying $p_{travel}$.

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
