# OpenReview forum: "On Early Extinction and the Effect of Travelling in the SIR Model"
_auai.org/UAI/2022/Conference — UAI 2022 Poster_

### Official Review · Reviewer_BTt4 · 2022-04-08

**Q2(1) Originality/Novelty:** 3
**Q2(2) Significance/Impact:** 3
**Q2(3) Correctness/Technical Quality:** 3
**Q2(6) Clarity Of Writing:** 4
**Q6 Overall Score:** 8
**Q8 Confidence In Your Score:** 3

**Q1 Summary And Contributions:**

The paper discusses early extinctions and travel effects in a probabilistic population-based version of the SIR model. Early extinctions are defined as the situation where the number of infected people never exceeds $\sqrt{N}$, for a population of size $N$. The probability of early extinctions is tied to the reproduction number. The paper further investigates how travel affects the number of infections in two countries with different infection rates.

**Q2 Assessment Of The Paper:**

More detailed information regarding each of these aspects is given below:

**Q2(4) Quality Of Experiments (Optional):**

4: Excellent: The experimental evaluation is comprehensive and the results are compelling.

**Q2(5) Reproducibility:**

4: Excellent: Key resources (e.g., proofs, code, data) are available and key details (e.g., proof sketches, experimental setup) are comprehensively described for competent researchers to confidently and easily reproduce the main results.

**Q3 Main Strengths:**

The paper is very well written and clearly structured, and the experiments are instructive and insightful. Although I have not checked the proofs carefully, the paper appears sound. The simulation of two countries is interesting and suggests further work towards proving Conjecture 1 (even if just for a simplified system of coupled differential equations).

**Q4 Main Weakness:**

I do not see major weaknesses in the paper, except for the fact that Conjecture 1 is not proved yet. I would expect that using a system of coupled differential equations, some supportive statements can be derived, but I understand if this is out of scope of the paper.

**Q5 Detailed Comments To The Authors:**

- At some point, the notation is not fully clear. Not every reader (including myself) is so familiar with Landau notation such that, e.g., $I_{t*}\in[\omega(1),o(N)]$ is readily understood. Please consider complementing these statements either with a definition, with a verbal explanation, or with both.
- What does $\sim$ mean in (1)? Approximation ($\approx$) or asymptotic behaviour?
- In Th. 2, the expected total number of infections is sublinear. I assume that this means sublinear in $N$. Does this not imply that the total number of infections is $o(N)$ instead of $o(1)$?
- In Th. 3 and 6 it is claimed that a quantity is concentrated around its expected value. This could be stated more precisely.
- On top of page 5, a reference to an equation is unresolved ("(1)-(1)").
- In the enumeration on page 6, the usage of $S$ in point 3 is confusing. Is this $S$ related to $S_t$? Please explain. Also, in this context, the word "couplings" may be confusing, as it is loaded with a different meaning in stochastics.
- Footnote 2 has some problems: "infection rates" should be the reproduction number, "rations".
- In Sec. 5 you set $\gamma=0,2$, does that mean $\gamma=0.2$? Also, you set $\beta^1 = 0 \le \beta^2=0$. While the statement is correct, I assume that there must have been a mistake.
- The figure legends and axis labels should be larger.

**Q7 Justification For Your Score:**

The paper discusses an interesting topic from a (to the best of my knowledge) novel perspective. It is well written and clear.

**Q9 Complying With Reviewing Instructions:**

1: Yes.

---

### Official Review · Reviewer_CBUJ · 2022-04-10

**Q2(1) Originality/Novelty:** 3
**Q2(2) Significance/Impact:** 2
**Q2(3) Correctness/Technical Quality:** 3
**Q2(6) Clarity Of Writing:** 3
**Q6 Overall Score:** 6
**Q8 Confidence In Your Score:** 2

**Q1 Summary And Contributions:**

Authors develop several algorithms to analyse the outcome of a pandemic where they model each individual with three states and they only assume pairwise interactions. They also empirically analyse what happens if individuals travel across two counteries and observe the surprising result that very large travel rate may decrease the total number of infected individuals. They propose a conjecture based on their empirical findings.

**Q2 Assessment Of The Paper:**

More detailed information regarding each of these aspects is given below:

**Q2(4) Quality Of Experiments (Optional):**

3: Good: The experimental evaluation is adequate, and the results convincingly support the main claims.

**Q2(5) Reproducibility:**

4: Excellent: Key resources (e.g., proofs, code, data) are available and key details (e.g., proof sketches, experimental setup) are comprehensively described for competent researchers to confidently and easily reproduce the main results.

**Q3 Main Strengths:**

The paper seems to make a significant theoretical contribution to disease modeling in a population.

**Q4 Main Weakness:**

I don't see any major weaknesses but I am not an expert in this field and might be missing some relevant literature.

**Q5 Detailed Comments To The Authors:**

The paper definitely delivers what it promises to. However I am not an expert in this field and cannot judge the novelty and importance of the specific settings in the context of the existing literature. I am reflecting this in my confidence score.

Could you comment on why the interactions are restricted to pairs of nodes? We know that superspreader events typically involve an individual infection several individuals at the same time. Could this be incorporated?

Is there a fundamental reason for choosing sqrt(N) for extinction? Does it really go extinct if this is not reached in a certain time? Calling reaching sqrt(N) an extinction is quite confusing and makes interpreting the theoretical results difficult.  [I realize this is later proven, which is great. But please mention it here early on as well to clarify right there why this choice is made.]

large p_travel can push total infections below no-travel case. This is definitely interesting but its significance is diminished if one incorporates the large error bars at p_travel=0. Could you comment on this?

r_infinity is concave in R0 is mentioned in the experiments section. Authors conduct an experiment to show this. Is there a way to show this claim theoretically>?

The explanation for the conjecture seems convincing that beta's might "mix together".

Minor comments:
- allowing which allows
- by coupling them which each other

**Q7 Justification For Your Score:**

I don't see a reason to reject the paper but it is hard to assess the impact for me. Therefore I recommend acceptance but with low confidence.

**Q9 Complying With Reviewing Instructions:**

1: Yes.

---

### Official Review · Reviewer_U36Y · 2022-04-12

**Q2(1) Originality/Novelty:** 2
**Q2(2) Significance/Impact:** 1
**Q2(3) Correctness/Technical Quality:** 3
**Q2(6) Clarity Of Writing:** 2
**Q6 Overall Score:** 2
**Q8 Confidence In Your Score:** 3

**Q1 Summary And Contributions:**

The paper studies the SIR model for modeling the spread of disease. The probability of early extinction of the disease (as opposed to becoming pandemic) is studied in one-country and two-country cases. Besides, the effect of travel on the number of infections is studied.

**Q2 Assessment Of The Paper:**

More detailed information regarding each of these aspects is given below:

**Q2(4) Quality Of Experiments (Optional):**

2: Fair: The experimental evaluation is weak: important baselines are missing, or the results do not adequately support the main claims.

**Q2(5) Reproducibility:**

2: Fair: Key resources (e.g., proofs, code, data) are unavailable but key details (e.g., proof sketches, experimental setup) are sufficiently well-described for an expert to confidently reproduce the main results.

**Q3 Main Strengths:**

The paper studies a problem related to a topic of wide concern in recent years.

**Q4 Main Weakness:**

I am concerned that this paper does not fit into the scope of UAI.

**Q5 Detailed Comments To The Authors:**

I would encourage the authors to send this work to venues focused on epidemiology, in particular, on mathematical modeling of infectious diseases, for proper evaluation and wider reception.

**Q7 Justification For Your Score:**

I do not think this paper is related to "AI" (loosely defined) or particularly relevant to the subject areas in https://www.auai.org/uai2022/subject_areas.

**Q9 Complying With Reviewing Instructions:**

1: Yes.

---

### Decision · Program_Chairs · 2022-05-15

**Decision:**

Accept (Poster)

**Comment:**

Meta Review: The papers does as advertised: an analysis of a variant of the popular SIR model, including the modeling of multi-country interaction. It's a result interesting in itself, particularly if we reflect on how much the SIR models were referred to during the early stages of the covid pandemic. Concerning whether it fits the conference, I believe it formally fits well, as stochastic processes and probabilistic relational models have been an integral part of UAI.  That been said, I think the authors would increase the exposure of their work by targeting an applied probability or epidemiology journal.